# Biomaterials for Cell-Surface Engineering and Their Efficacy

**DOI:** 10.3390/jfb12030041

**Published:** 2021-07-13

**Authors:** Seoyoung Jang, Jin Gil Jeong, Tong In Oh, EunAh Lee

**Affiliations:** 1Department of Medical Engineering, Graduate School, Seoul Campus, Kyung Hee University, Seoul 02447, Korea; syjang329@khu.ac.kr (S.J.); scvjjhg@khu.ac.kr (J.G.J.); 2Department of Biomedical Engineering, School of Medicine, Seoul Campus, Kyung Hee University, Seoul 02447, Korea; tioh@khu.ac.kr; 3Impedance Imaging Research Center, Seoul Campus, Kyung Hee University, Seoul 02447, Korea

**Keywords:** stem cells, encapsulation, cell therapy, cyto-protective effect, cell surface modification

## Abstract

Literature in the field of stem cell therapy indicates that, when stem cells in a state of single-cell suspension are injected systemically, they show poor in vivo survival, while such cells show robust cell survival and regeneration activity when transplanted in the state of being attached on a biomaterial surface. Although an attachment-deprived state induces anoikis, when cell-surface engineering technology was adopted for stem cells in a single-cell suspension state, cell survival and regenerative activity dramatically improved. The biochemical signal coming from ECM (extracellular matrix) molecules activates the cell survival signal transduction pathway and prevents anoikis. According to the target disease, various therapeutic cells can be engineered to improve their survival and regenerative activity, and there are several types of biomaterials available for cell-surface engineering. In this review, biomaterial types and application strategies for cell-surface engineering are presented along with their expected efficacy.

## 1. Introduction

Since it was discovered that various regenerative cell sources can exert essential functions to promote the regeneration of tissues damaged by injury or disease, the number of clinical applications of cell therapeutic agents has continued to increase. Although many cell-based therapies can provide valuable functionality that cannot be exerted by any other therapeutic options, the duration of the efficacy could be limited due to the fact that an in vivo environment is not favorable for the survival of the transplanted cells. Cell encapsulation can increase the duration of cell survival by providing protection from unfavorable environments.

Cell-surface modification or encapsulation technology was originally developed with microbes such as yeasts [1] and *Escherichia coli* [2]. Various materials such as silica gel [3], poly(styrene sulfonate sodium salt), poly(allylamine hydrochloride) [1], chitosan, alginate, hyaluronic acid [2], poly(diallyl dimethylammonium chloride) [4], and silk [5] were adopted for encapsulation of microbes. However, to apply the cell encapsulation technology for mammalian cells, the biocompatibility of the encapsulation process and its effect on cell characterization should be considered.

There are several previous reports on mammalian cell encapsulation, which show diversity in the approaches and the biomaterials used. Such diverse cell encapsulation technologies could be classified into several categories based on cellularity (single-cell/multi-cellular) and the dimension of encapsulation thickness (nano-encapsulation/micro-encapsulation) [6]. In each category, the extent of cell–cell contact and cell–ECM interaction can vary based on the encapsulation method and the characteristics of the biomaterials (Figure 1).

The ultimate purpose of mammalian cell encapsulation is to maximize the survival and functionality of the cells for therapeutic purposes. Matrix molecules incorporated in the encapsulation hydrogel binds to matrix receptors on the cell plasma membrane surface and activates a signal transduction pathway leading to Akt protein phosphorylation—the marker for cell survival activation—which exerts an inhibitory effect on apoptosis induction [7,8].

Components and methods of cell encapsulation have been developed to meet the challenges posed in the application for target therapeutic purposes. The functionality of systemically transplanted MSCs (mesenchymal stem cells) administered through intravascular injection might be potentiated through single-cell nanoencapsulation [7,9]. For the mass delivery of therapeutic cells, multi-cellular encapsulation has been adopted. For example, the intended therapeutic activity of islet β-cells is the production of insulin. To stabilize and secure the activity of the enzyme production, islet β-cells were micro-encapsulated in a multicellular state [10].

The literature reporting the promising outcome obtained from multicellular micro-encapsulation indicates that the cell encapsulation technology exists right next to the borders of tissue engineering. The technologies developed for cell encapsulation can benefit tissue engineering, while also working the other way around. However, cell encapsulation technology covers a broad area, from the scale-up preservation of cells to its therapeutic application. Encapsulated cells could be utilized for tissue engineering by upstream production and application. Cell encapsulation can serve not only a tissue regeneration purpose, but also specific therapeutic purposes involving immune cells [11].

As much as the subjected cells are varied, the methods of cell-surface engineering are diverse. Functional small molecules could be tethered on the cell surface by covalent conjugation methods such as chemical conjugation, enzymatic/metabolic conjugation, physical conjugation, or UV-mediated grafting. Several review papers have covered diverse techniques adopted for cell-surface engineering [12,13,14]. In this review, current literature reads for non-covalent modification of mammalian cell encapsulation are summarized with a focus on the type of biomaterials employed for mammalian cell encapsulation, the types of cells, and the target efficacy that can get benefit from encapsulation. At the end of this review, future perspectives are briefly discussed.

## 2. Methods of Cell-Surface Engineering and the Type of Biomaterials

Literature indicates that various biomaterials have been used for cell-surface engineering, and the adopted method depends on the characteristics of the employed biomaterials, supplemented components, and target efficacies. The surface-engineering methods according to the type of biomaterials are summarized in Table 1.

### 2.1. Self-Assembly

#### 2.1.1. Self-Assembly of Single Components

Most of the matrix molecules have a repeated structural motive that facilitates intra-molecular assembly or inter-molecular assembly. Type I collagen is the typical example in this category. Collagen molecules exist in a well-dissolved solution state in an acidic environment. When the pH and temperature are adjusted to the physiological range, collagen molecules can self-assemble to form a gel-like state. Collagen is the most abundant matrix component in the body and can provide a cell-friendly growth environment. The self-assembly of a single component provides a stable mechanism that is highly feasible to produce cell encapsulation with an encapsulation layer with a micro- to macro-scale thickness.

#### 2.1.2. LbL Assembly

A layer-by-layer (LbL) assembly technique includes the deposition process of flexible films with a nano-thickness. The mechanism of layer deposition is based on the affinity between the materials of the new layer and the previous layer. Many cases involve electrostatic interaction between alternating layers (Figure 2). While LbL assembly-based surface modification produces a highly flexible nano-thin film on the cell surface such that the applied layer does not strain the cells’ normal activity, the deposition of multi-layer alginate, chitosan, poly-L-lysine, and polyethylene glycol (PEG) by LbL assembly successfully modified the membrane surface of red blood cells (RBCs) and endowed ‘immunocamouflage’ RBCs.

### 2.2. Cross-Linking

The very first functional recovery with encapsulation technology was tried in 1980 by Lim et al. by an encapsulation of pancreatic islet tissues with alginate. The gelation of alginate involves the chelation of multi-valent cations to carboxylic acid groups on alginate polysaccharide backbone, which is composed of [1,2,3,4]-linked blocks of beta-D-mannuronate and alpha-L-guluronate, and Ca^2+^ is the most frequently used divalent ion for the gelation of alginate. Being optically clear, the behavior of the encapsulated cells can be easily observed. Through the gelation process, alginate produces a thick and robust layer such that the resultant encapsulated cell/tissues are very stable for handling, and it protects from exposure to immune cells’ recognition (Figure 3). Therefore, not only islet beta-cells but other types of cells, including hepatocytes, endothelial cells, neuronal cells, and MSCs, were encapsulated in alginate and resulted in a cytoprotective effect, an improved cell activity/metabolic index, and protection from the immune response.

### 2.3. Polymerization

Polymerization conditions need to meet certain criteria to be adopted for cell-surface engineering. The range of thermal conditions, reactants, and a by-product of the reaction need to be physiologically safe. In addition to this, it is desirable that the product of the polymerization produces a network-like macromolecule, because the chain-like molecule is not effective in covering the cell plasma membrane surface. The poly-dopamine (PDA)-based surface modification of RBCs induced the masking of cell-surface antigenic epitopes and resulted in 100% survival in an in vivo study, even with repeated transfusion.

## 3. Therapeutic Cells Subjected to Surface Engineering

### 3.1. BM-MSCs

BM-MSCs have immunomodulation characteristics and therefore were used for the treatment of various immune-related diseases, such as graft versus host disease (GvHD), sepsis, and therapies for autoimmunity disorders (such as type 1 diabetes and Crohn’s). The mode of action includes communication by cell–cell interaction and paracrine signaling (secretion of soluble factors) [46]. BM-MSCs administered to an experimentally established GVHD model showed massive programmed cell death. Although BM-MSC-mediated immunosuppression can be exerted through a contact-independent mechanism [47], a significant number of cells still need to survive after in vivo administration to exert their function. BM-MSCs were proven as promising therapeutic options to acute GVHD when they do not respond to steroid regimens [48,49]. Through the same mechanism, BM-MSCs also ameliorate the pathological symptoms of sepsis, which involves whole body inflammation caused by blood injection [50]. Other studies have provided evidence that BM-MSCs can not only attenuate the severity of the end-organ injury but also effectively ameliorate septic coagulopathy, alleviate vascular damage, reduce inflammation, attenuate acute lung injury, and improve the survival rate [51,52]. As shown by recent clinical trials, osteoarthritis patients injected with in-vitro-expanded BM-MSCs showed pain reduction and the recovery of functionality [53]. The efficacy of BM-MSCs’ regenerative therapy is based on the trophic effect that stimulates neighboring parenchymal cells to start repairing damaged tissues, and its target diseases include ischemic stroke, arterial disease, and myocardial infarction [54]. However, IV-injected MSCs barely reach the intended target tissue because they are exposed to high shear stress in circulation, which results in poor cell survival. Cells encapsulated with a dextran-based hydrogel and alginate show better survival and cell stabilization [32,36].

Surface-engineered MSCs show better cell survival. BM-MSCs encapsulated with hydrogels such as dextran or alginate showed maximum stability in cell survival since such hydrogel-based encapsulations result in thick and stable hydrogel layers around the cells. However, a thick encapsulation wall results in an increased diameter of the encapsulated cell composite and increases the risk of vein occlusion upon IV injection, thereby excluding the possibility of systemic injection. [32,36] 

Surface modification of BM-MSCs by LbL assembly using PLL/HA resulted in nano-thin films around the cell plasma membrane surface. LbL-assembled layers resulted in a 20-nm-thick discontinuous and patch-like structure. Although this structure seemed unsturdy, both in vitro and in vivo cell survival was significantly potentiated, and improved recruitment to the wound site was observed in the muscle injury animal model (Figure 4) [7,9,55].

### 3.2. Islet Beta-Cells

Diabetes patients suffer from a lack of control over glucose metabolism caused by insulin insufficiency. Type I diabetes is caused by autoimmune-mediated beta-cell destruction, and the current treatment regimen for diabetes relies on the administration of exogenous insulin by frequent injections. Natural islet beta-cells sense blood glucose levels and respond by producing insulin for exactly the right duration to control blood glucose levels within a strict physiological range, while exogenous insulin delivery often fails to control blood glucose levels. Pancreatic islet transplantation can be an effective treatment for achieving naturally controlled insulin production in response to blood glucose levels.

In experimental conditions, the xenogenic transplantation of hESC-derived β-cells into mice ameliorated hyperglycemia in diabetic mice [56]. After the transplantation, human C-peptide was detected until 8–12 weeks upon transplantation, which is significantly shorter compared with the transplantation of pancreatic progenitors (20 weeks), yet longer compared with the transplantation of cadaveric islets (2 weeks) [57,58].

Allogeneic beta-cell transplantation might provide a functional restoration of islet beta-cells and a promising therapy for type I diabetes. However, allogeneic cell transplantation necessitates life-long immune suppression, which might cause graver side effects. [45]. To solve this problem, cell encapsulation was adopted for beta-cell transplantation. A sound and robust layer formed by micro-encapsulation can serve as a barrier from aggressive immune cells’ attack.

Microencapsulation with high-mannuronic-acid alginate allowed for the prolonged survival of allogeneic transplanted islets in diabetic mice [27,28]. While the micro-encapsulation of islet beta-cell provides protection from the immune system (immune cells and antibodies), nutrients from the body and the insulin proteins produced by islet beta cells need to pass through the encapsulation for the transplanted beta-cells to prolong that function. These two points are major remaining challenges for the successful encapsulation of islet beta-cells.

### 3.3. Endothelial Cells

Diseases associated with the vascular system, particularly those associated with cardiac and cerebrovascular disease is the leading cause of death worldwide. The current therapeutic approach includes pharmacological treatment that can involve side effects and surgical revascularization. Because these approaches cannot completely reverse pathophysiology, many experimental therapeutic approaches utilize endothelial progenitor cells (EPCs) to treat cardiovascular diseases (CVDs). Active EPCs served as a potent source of the essential mediators of a new vessel formation [59].

In a murine model of peripheral limb ischemia, human EPC injections improved tissue reperfusion, and limb salvage rate was significantly increased [60]. In clinical trials, the injection of EPCs to idiopathic pulmonary hypertension patients showed significant improvement in terms of pulmonary artery pressure, pulmonary vascular resistance, and cardiac output [61].

However, a low retention rate due to cell scattering after injection was still a major concern. To stabilize cell retention and survival, various approaches were employed with EPC transplantation. Hyaluronan (HA) was known to increase the proliferation of human umbilical vein endothelial cells (HUVECs) and protect apoptosis. Injecting HUVECs with HA showed a significantly improved outcome in restoring blood perfusion and salvaging the ischemic limb when compared with HUVECs injected without HA [62].

The use of micro-encapsulation provided a cyto-protective effect to endothelial cells. When the elements of the encapsulation were composed to exert multifunctionality, encapsulated cells proliferated in a time-dependent manner and eventually released from the encapsulation and migrated to a nearby site in longer time points. The injection of multifunctional microgels encapsulating endothelial cells and growth factors induced neovascularization in animal models of hindlimb ischemia [29,63].

### 3.4. Hepatocyte

Although liver tissue shows the great regenerative capacity and spontaneous recovery upon mild damage or partial hepatectomy, acute/chronic liver failure, metabolic disorders such as Crigler-Najjar syndrome type 1 (CN-2), or urea cycle defects with hepatic basis can be a life-threatening situation, and organ transplantation is needed [64].

If cell therapy could replace organ transplantation, it could be a better therapeutic option because cell therapy involves a less invasive transplantation procedure, and the cryo-preserved cell source makes it immediately available in an emergency [65]. An in vitro study showed that hepatocytes encapsulated using alginate-poly-L-lysine-alginate showed a higher albumin secretion at 2 weeks of culture when compared with non-encapsulated free primary hepatocytes. The transplantation of such encapsulated hepatocytes increased the survival of mice. The encapsulation technique even exerted a cyto-protective effect against cryo-injury [22,24]. Thick hydrogel layer-encapsulation can exert a beneficial role by holding and slowly releasing paracrine factor, which can be essential for a hepatocyte’s survival. In recent experimental studies, hepatocytes encapsulated in alginate microbeads were transplanted to a mouse acute liver failure model and resulted in improved survival [19,20,21].

However, the same challenges exist as in the case of islet beta-cell encapsulation. The immunosuppression regimen to evade immune rejection is recognized as a high-risk factor for patients with coagulopathy. Additionally, a sufficient amount of target area needs to be secured for safe engraftment and further growth of the organ, which will be essential for patients’ long-term survival. Therefore, further research is needed to meet those challenges for in vivo transplantation and practical utilization in clinical situations.

### 3.5. Neuronal Progenitor Cells

Neural progenitor cells (NPCs) or neural stem cells (NSCs) are stem cells that can differentiate into the major cell types of the central nervous system. The potential of NPCs to directly replace damaged tissue make it a promising therapeutic cell type for many neurodegenerative diseases such as Parkinson’s disease, Alzheimer’s disease, or spinal cord damage.

Although NPCs have high therapeutic potential, they show an inferior survival rate upon transplantation. Additionally, the biochemical and biomechanical environment is important for retaining their functionality, such that neuronal stem cells (NSCs), including NPCs, are maintained in a spheroid culture to retain their differentiation potential. Especially, matrix components generated by cells during spheroid culture induce three-dimensional cell–cell interactions and cell–ECM interactions, which play important roles in cells’ long-term survival and stemness [39]. When NPCs were encapsulated by a 3D network of nanofibers containing laminin epitope (isoleucine-lysine-valine-alanine-valine, IKVAV) on their surface, the cells rapidly differentiated into neurons in a laminin or soluble peptide-dependent manner [6]. As for their biomechanical property, NPCs cultured in 3D hydrogels with higher degradability showed proliferation three days after the encapsulation, while NPCs encapsulated in a low degradable hydrogel did not show proliferation over two weeks. Both the degradability and remodeling time exerted by the hydrogel encapsulation significantly impact NPCs’ differentiation capacity [37]. Therefore, developing an encapsulation technique with the right biomaterial components for NPCs might result in excellent long-term viability and functionality.

## 4. Functional Aspect of Cell Encapsulation

### 4.1. Protection from Apoptosis

#### 4.1.1. Protection from Cell Death Caused by Mechanical Stress

Animal cells in culture are exposed to various stresses coming from the culture environment and are susceptible to damages that eventually lead to cell death. Among these, mechanical stress can cause direct damage to the cultured cells. In mild degrees, mechanical stress can activate cells to survive [66] or express mechano-transduction-related genes and facilitate the differentiation towards certain types of tissues that undertake mechanical functions [67]. However, extended exposure to mechanical stress to a high extent during the handling process or cultivation can cause cell death.

Even simple centrifugation could be critically detrimental for mechanical-stress-sensitive cell types such as hepatocytes or neuronal cells. A study with hepatocyte carcinoma showing extremely high cell death after four rounds of centrifuge indicates that hepatocytes are highly susceptible to mechanical stress [8]. Cell culture in the 3D rotary vessel can be an in vitro model that replicates the situation of blood vessels in vivo with high shear stress. MSCs in attachment-deprived states showed high susceptibility to shear stress in vitro [68] and in vivo [7].

The high susceptibility to mechanical stress is based on the fact that the lipid bilayer of the cell plasma membrane is constructed by a hydrophobic interaction between phospholipid molecules. However, hepatocytes, highly susceptible to mechanical stress in an attachment-deprived state, show mechanical stability inside the liver tissue surrounded by matrix molecules. Likewise, cell surface modification by the deposition of hydrogels or matrix molecules can dramatically reduce mechanical insult by providing mechanical stability. When HepG2 cells (hepatocyte carcinoma) were coated with FN-gelatin or Col IV-LN by LbL assembly, more than 85% of the cells survived after nine rounds of centrifugation, while the non-coated control group showed poor cell survival (6%) [8]. Furthermore, the application of high-mechanical strength molecules can enhance mechanical stability even further. In a study reported by Cha, cells seeded on a microcarrier-like gelatin core were encapsulated with silica. The silica encapsulation protected the cells from mechanical stress and resulted in significantly increased viability [40]. Systemic transplantation to in vivo animal models also proved that surface-modified cells with LbL assembly resulted in better cell survival [7,15,69].

#### 4.1.2. Anoikis-Preventive Effect

When cells make attachments to the substrate, ECM molecules bind to their corresponding receptor and transduce survival signals to the cells. Therefore, when the cells are detached from the stroma and are left in the attachment-deprived state for a certain duration, the cells eventually undergo anoikis, a specific type of apoptosis caused by an attachment-deprived state. Most cell surface-located matrix receptors such as integrin or CD44 can activate survival signaling pathways leading to Akt activation. Therefore, cell encapsulation mimicking the interaction between matrix molecules and matrix receptors on the cell surface can result in Akt activation and significantly increase the survival of cells in the attachment-deprived state. When BMSCs were surface-modified by LbL assembly composed of poly-L-lysine and hyaluronic acid, the cell survival rate and Akt activation were significantly increased [2]. When the BMSCs were surface-modified with LbL assembly composed of type I collagen and hyaluronic acid, the cell surface index, which indicates the openness of cell plasma membrane surface, showed that cell surface openness was not significantly decreased even with eight layers of LbL assembly. This indicates that the matrix deposition might be localized to the matrix receptor area. The thickness the local deposition induced by eight layers of LbL deposition was roughly 20–30 nm. Discontinuous matrix patches with a 20–30 nm thickness localized to matrix receptors are not likely to exert mechano-transduction signaling. Therefore, it is suggested that the interaction between the cell surface and LbL-assembled matrix molecules was based on the intracellular survival signaling pathway, which was enough to significantly increase the duration of cell survival [3].

#### 4.1.3. Cryopreservation

During the cryopreservation of animal cells, a cryoprotective agent (CPA) such as DMSO was used to suppress the damage caused by ice crystal formation. Although there is some controversy about the active concentrations, the CPA has a negative effect on cell activity-inducing proliferation arrest or cytotoxicity. Many reports show that a CPA such as DMSO results in reduced proliferation at a concentration of around 1–2%, and a concentration over 10% causes the cell number to decrease due to cytotoxicity [70]. However, during the cryopreservation of animal cells, ice crystal formation results in damages of cell plasma membrane integrity, and this destructive effect is greater compared with the damage caused by CPA. Additionally, cellular membrane damage can lower cell recovery upon thawing. Porcine adipose-derived stem cells (ADSCs) are also highly susceptible to the damage caused by ice formation during the cryopreservation process. When the porcine ADSCs were encapsulated with alginate shell structures, damage caused by cryopreservation decreased 3.5-fold, even with a lower concentration of the CPA (Figure 5) [33].

The cryopreservation of encapsulated cells might be utilized for the preservation of therapeutic cells with transplantation purposes. Alginate-encapsulated hepatocytes induced promising outcomes when they were intra-peritoneally transplanted to an acute liver failure animal model, which is a life-threatening condition without liver transplantation [10,23]. As acute liver failure needs emergency transplantation, the cryopreservation of allogeneic hepatocytes in a transplantation-ready formulation has great advantages. Rat hepatocytes triple-encapsulated with an alginate–chitosan–alginate shell exerted a normal liver function after thawing from cryopreservation both at −80 °C and in liquid nitrogen, which was confirmed by the CYP450 activity [25]. Likewise, further target diseases and therapeutic purposes could be found for various types of cryopreserved encapsulated cells. 

### 4.2. Protection from Immune Rejection

The development of cell therapeutic agents is intended for a pathological situation where ordinary drug-based therapy fails to cure, and it is highly likely that the cell sources from the highly pathological site cannot yield promising effects for regeneration. In the case of type I diabetes, islet β-cells in the pancreas cannot produce enough insulin to metabolize glucose. To avoid frequent injection of insulins and find radical treatment, the possibility of allogeneic β-cell transplantation has long been investigated [26]. For allogeneic cell therapy, protection from immune rejection is especially important because immune rejection can damage transplants and compromise the function of transplanted cells. Studies with animal diabetes models showed that the encapsulation of aggregated multicellular islet β-cells by polycaprolactone protected from immune rejection without immune-suppressive agents [4,45]. The biomaterial-based encapsulation for islet β-cells was designed to comprise nanopores to obtain nutrition for cell survival but block the access of host antibodies and immune cells [45]. Multi-layer polyelectrolyte-based nanoencapsulation also blocked the recognition by the host antibody and showed a comparable amount of insulin release when such cells were compared with non-encapsulated cells [4].

Not only for stroma-attached islet β-cells, but also circulating blood cells such as red blood cells (RBCs) were surface-modified to evade host immune rejection or blood coagulation induced by antibodies against the corresponding ABC type [34]. Poly-dopamine adopted for surface modification of human RBCs did not compromise the cell functions and showed a normal survival profile when they were injected into a murine animal model, indicating a perfect immune-protective effect, even for an individually scattered single-cell state [38].

### 4.3. Modulation of Cell Growth and Hatching

In some studies, encapsulated cells showed no increase in cell number during in vitro culture periods. During four days of culture, HeLa cells and human BM-MSCs encapsulated in an enzyme-responsive polymer nano-shell maintained a surviving state without cell proliferation, while non-incapsulated HeLa cells and BM-MSCs showed an approximately 10-fold and 8-fold increase in cell number, respectively [9]. In other cases, proliferation persists for a limited duration of time if the encapsulation is stiff and do not allow structural remodeling that is needed to hold an increased number of cells [30,35]. Cells clustered inside the encapsulation cause cell–cell contact, which results in potentiated cell survival and higher mitotic activity. If the biomaterial comprising the encapsulation has a lower degree of cross-linking, cells proliferate within the encapsulation and cause a greater aggregation of cells [31,35]. This shows that cells can sense the mechanical milieu of the environment, and the expandability is exhibited accordingly. Based on these mechano-transduction-induced cell characteristics, the cell proliferation state during the encapsulation state can be controlled by the stiffness of the encapsulation.

When encapsulation is not stiff enough to repress cell expansion and refrain the cell activity, the encapsulated cells hatch. The incorporation of hyaluronic acid in alginate-based encapsulation provided a softer spot in the encapsulated structure, and the cells hatched out of the encapsulation, while the cells encapsulated in the alginate bead stayed inside during the other culture [35]. The incorporation of gelatin into the agarose-based encapsulation provides the degradation site because gelatin can be degraded by enzymes produced by the cells. In proportion to the gelatin content, the release of encapsulated cells increased at all time points [42]. The hatching or release of the encapsulated cells might be beneficial to the therapeutic outcome after the transplantation.

When MSCs encapsulated in hydrogel composed of agarose, dextran sulfate, and collagen were transplanted to the myocardial infarction site, encapsulated cells were slowly released and integrated into adjacent tissues. This slow release of cells showed a beneficial effect in various aspects. The released cells easily integrated into adjacent tissues when compared with cells transplanted in an attachment-deprived single cell state. The gradual integration of released cells proceeded without provoking abrupt changes or a foreign body response, resulting in active participation in the wound healing process in the myocardial infarction site [34].

## 5. Future Perspectives

### 5.1. Conrol of Cell Activity

According to the encapsulation strategy and the type of employed biomaterials, the activity of cells in encapsulation can be controlled. Cells could remain stable without significant changes of the proliferative state if stiff biomaterials were adopted for encapsulation [9,21,42]. Based on the report showing that nano-scale LbL assembly could keep the encapsulated cells dormant for at least eight days, the cell encapsulation does not need to exhibit a micro-scale thickness to achieve cellular activity control [9]. 

Basically, stem cells residing in the body are close to the state of being dormant and non-differentiated. The signals coming from the environment activate the stem cells to proliferate and get differentiated to show functionality. Many previous studies have shown that those environmental signals can be a chemical moiety [71] or a mechanical property [5,72] and alter the stem cells’ characteristics and steer the path of cells’ functionality.

However, cells for a therapeutic purpose are subjected to scrutiny for their safety and stability in their functionality. Therefore, the characteristics of encapsulated cells during dormancy and after hatching need to be defined in detail.

### 5.2. Tissue Engineering from Microscale

Classical approaches of tissue engineering include cell seeding to a porous scaffold of a desired shape and volume. If the target tissue for therapy is independent of formulating shape and dimension, encapsulated cells can serve as a good strategy for the recovery of tissue functionality. The transplantation of islet beta-cells and hepatocytes in their encapsulated stats can be good examples.

As cell spheroids or aggregates are increasingly used as a building block for the tissue engineering strategy [7,73], encapsulated cells can also be utilized as building blocks for three-dimensional tissue construction. Taking into consideration that the encapsulation of cells might be categorized based on single-cellular/multicellular and nanoencapsulation/microencapsulation according to the state of cells and their interaction with the surrounding materials [1], encapsulated cells have advantageous traits in the sense that the surface of each building block is available for further modification and that the design of materials can endow and modulate the subtle characteristics of cells.

When designing cell encapsulation, the conditions required for the functionality of the target tissue should be considered. In the case of reconstructing cartilage tissue, maintaining cell–cell contact is critical [41]. Flexible nanoencapsulation can permit cell–cell contact between encapsulation units, but if the encapsulation thickness increases, cell–cell contact between different encapsulation units cannot be made, which results in insulation between encapsulated beads. By introducing the degradation site into the backbone of encapsulation, the insulation can be resolved in a predictable fashion. Likewise, the hatching of cells could be controlled by introducing an appropriate degradation motif of “weak points” so that the hatching and cell release takes place in the intended environment or timing.

## 6. Conclusions

The clinical application of cell therapy can save repetitive drug injections or acute symptoms, which can be exemplified by type I diabetes or graft-versus-host disease (GVHD), respectively. In the case of injected cells, the modulation of cell activity after the injection is beyond control. Therefore, stabilizing cell quality even after the injection or transplantation can result in augmented therapeutic activity. The ultimate goal of developing technology for potentiated cell therapy is to be utilized for human welfare. Therefore, a clinical trial of an encapsulated cell therapeutic agent needs regulatory approval, which requires scrutiny for its safety and therapeutic effects. The complexity of the method can result in challenges in the scale-up manufacturing of the therapeutic agent. Likewise, the complexity of the therapeutic agent leads to regulatory complexity [74]. Therefore, for the practical application of cell surface modification and encapsulation, it is important to design the method that is scalable, meets regulatory requirements, and is market-viable.

## Figures and Tables

**Figure 1 jfb-12-00041-f001:**
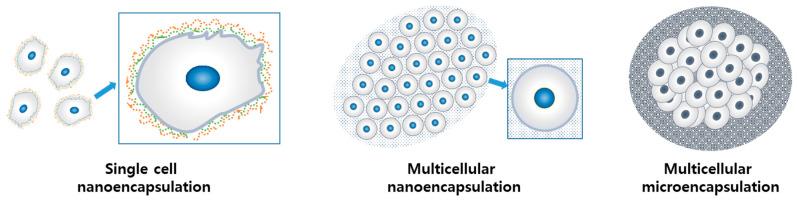
Three types of cell encapsulation according to the thickness of the encapsulation wall and cellularity. Reprinted and modified with permission from reference [6]. Copyright 2018 Wiley.

**Figure 2 jfb-12-00041-f002:**
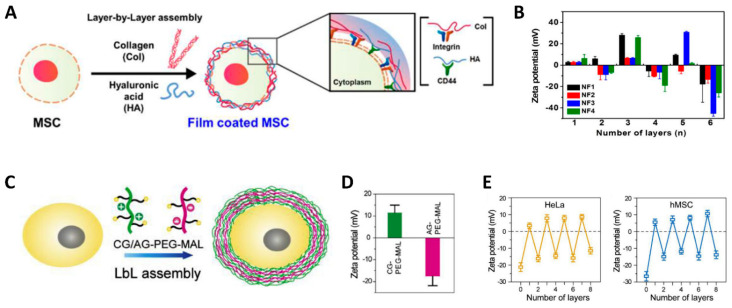
Schematic illustration indicating LbL assembly and the change in surface charge at each layer deposition. (**A**,**B**) Schematic illustration indicating the procedure of LbL assembly alternating Col I and HA deposition and surface charges after the deposition of each layer. Odd numbers indicate Col I deposition, and even numbers indicate HA deposition. Reprinted with permission from reference [9]. Copyright 2017 American Chemical Society. (**C**–**E**) Schematic illustration indicating the procedure of LbL assembly alternating cationic gelatin-PEG-maleimide (CG-PEG-MAL) and anionic gelatin-PEG-maleimide (AG-PEG-MAL). The surface charge of CG-PEG-MAL and AG-PEG-MAL was confirmed by Zeta potentila, and the surface charge of the cells during LbL deposition alternated between anionic and cationic states. Reprinted with permission from reference [18]. Copyright 2019 Elsevier.

**Figure 3 jfb-12-00041-f003:**
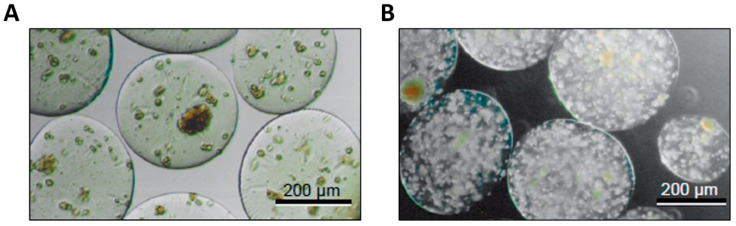
Hepatocytes encapsulated in alginate-chitosan-alginate microcapsules (**A**) and alginate microbeads (**B**). Reprinted after modification with permission from reference [25]. Copyright 2015 Taylor & Francis.

**Figure 4 jfb-12-00041-f004:**
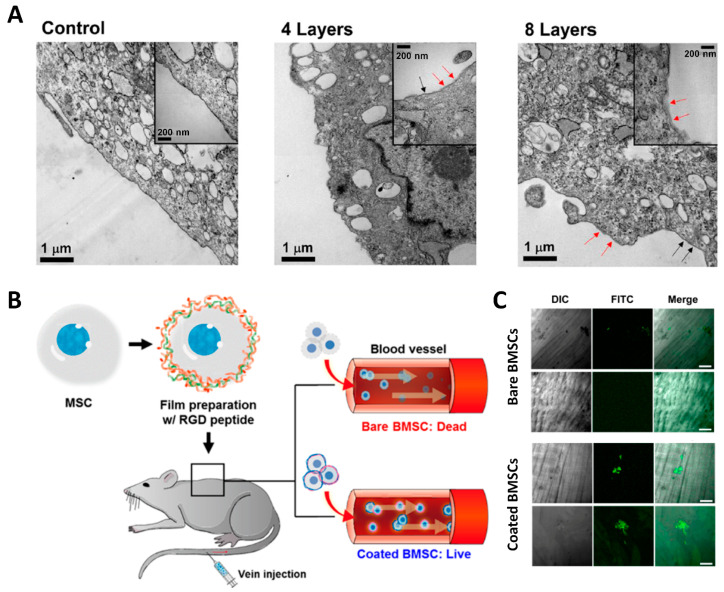
BMSCs surface-modified by LbL assembly. (**A**) TEM observation of LbL assembly composed of Col/HA deposition. The red arrows indicate the LbL assembly layers (an approximately 20 nm thickness), and the black arrows indicate a bare plasma membrane surface. Reprinted and modified with permission from reference [9]. Copyright 2017 American Chemical Society. (**B**) Schematic illustration indicating BMSCs with or without surface modification by LbL assembly subjected to systemic injection. (**C**) Surface-modified BMSCs injected after LbL assembly showed significantly higher recruitment to the wound site (White bar = 100 μm). Reprinted with permission from reference [7]. Copyright 2017 American Chemical Society.

**Figure 5 jfb-12-00041-f005:**
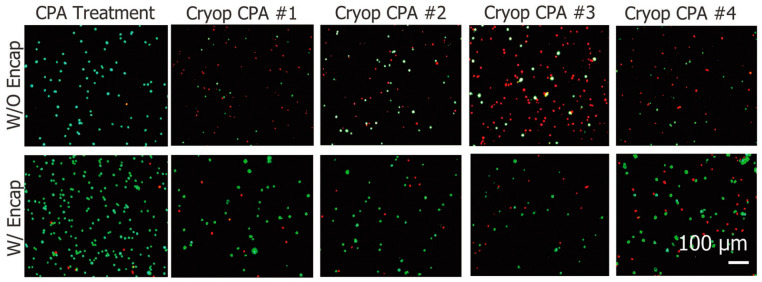
Cell encapsulation exerted a cyto-protective effect upon the cryo-preservation of ADSCs, as indicated by live/dead staining. Reprinted with permission from reference [33]. Copyright 2017 Wiley.

**Table 1 jfb-12-00041-t001:** Summary of cell-surface engineering techniques and the employed type of biomaterials.

Surface Engineering Methods		Materials	Cell Types	Proved Efficacy	Ref.
Self-assembly	Single component	IKVAV ^1^ peptide amphiphile	NPCs ^2^	Cyto-protective effect and better cell spreading/differentiation to more neuronal cells and fewer astrocytes	[15]
		Collagen/gelatin	Mouse fetal limb tissue	Single/clustered cell self-assembly observed during tissue development	[16]
	LbL assembly	HA ^3^, PLL ^4^	BM-MSCs ^5^, PBMCs ^6^, Hepatocytes	Cyto-protective effect by anoikis prevention, no interruption on cell activity	[7,11]
		PAH ^7^, PDADMAC ^8^, PSS ^9^	Islet tissue	Protection from immune response	[10]
		Collagen, HA	BM-MSCs	Cyto-protective effect, no significant decrease of surface index	[9]
		FN ^10^/gelatin or Col IV ^11^/LN ^12^	Hepatocytes (HepG2 cells)	Cyto-protective effect from mechanical stress	[17]
		PEG ^13^-gelatin	BM-MSCs, HeLa cells	Cytoprotective effect from enzymatic attack and mechanical stress	[18]
Cross-linking	Ionic cross-linking	Alginate	Hepatocytes	Improved liver metabolic index in acute liver failure, increased survival, cyto-protective effect in cryo-preservation	[19,20,21,22,23,24,25,26]
			Islet tissue, islet beta-cells	Survival of islet cells, secured insulin activity on glucose metabolism, protection from immune response	[26,27,28]
			Endothelial cells	Cyto-protective effect, neovasculogenesis	[29,30]
			Neuroblastoma cell line	Cell proliferation inside the capsule	[31]
			MSCs (BM, AD)	Long-term in vivo cell survival and cytokine production, cyto-protective effect during rapid-cooling cryo-preservation, long-term integration to the transplanted site	[32,33,34,35]
	Thio-Michael addition	Dex-GMA ^14^, DTT ^15^	BM-MSCs	Differentiation potential maintained	[36]
	Amine-reactive cross-linking	Elastin-like protein with adhesion/degradation domain	NPCs	Matric characteristics modulate the maintenance of NPCs differentiation potential, degradable matrix showed increase in neuronal marker expression	[37]
Polymerization	Chemical polymerization	PDA ^16^	RBC ^17^	Protection from immune response	[38]
	Photo-polymerization	Me-HA ^18^	iPSC ^19^-derived NPC	Stiffness of the matrix determines cells’ activity and survival, softer matrix produced better cell survival and tubule formation	[39]
		Me-gelatin ^20^	Cardiac side population cells	Protection from oxidative stress, mechanical stress, and immune response	[40]
		Me-PEG ^21^	BM-MSCs	Disruption of cell–cell contact by encapsulation showed negative efficacy in terms of chondrogenic potential	[41]
Etc	Combined method	Agarose/gelatin	BM-MSCs	Controlled release of encapsulated cells by gelatin%	[42]
		Alginate, Chitosan, PLL-PEG	RBCs	Protection from immune response	[43]
		Collagen, alginate, chondroitin sulfate, tannic acid, lignin	MSCs cell line	Potentiated osteogenic potential	[44]
	Macro-scale encapsulation	PCL ^22^	ES-derived beta-cell	Cytoprotective effect on islet cells, secured insulin activity on glucose metabolism, protection from immune response	[45]

^1^ IKVAV peptide: isoleucine-lysine-valine-alanine-valine peptide. ^2^ NPCs: neural progenitor cells. ^3^ HA: hyaluronic acid. ^4^ PLL: poly-L-lysine. ^5^ BM-MSCs: bone marrow-derived MSCs. ^6^ PBMCs: peripheral blood mononuclear cells. ^7^ PAH: poly-allylamine hydrochloride. ^8^ PDADMAC: poly-diallyldimethylammonium chloride. ^9^ PSS: poly-styrenesulfonate sodium salt. ^10^ FN: fibronectin. ^11^ Col IV: type IV collagen. ^12^ LN: laminin. ^13^ PEG: poly-ethylene glycol. ^14^ Dex-GMA: glycidyl methacrylate derivatized dextran. ^15^ DTT: dithiothreitol. ^16^ PDA: poly-dopamine. ^17^ RBC: red blood cell. ^18^ Me-HA: methacrylated hyaluronic acid. ^19^ iPSC: Induced pluripotent stem cells. ^20^ Me-gelatin: methacrylated gelatin. ^21^ Me-PEG: methacrylated poly-ethylene glycol. ^22^ PCL: poly-caprolacton.

## Data Availability

Not applicable.

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
