# Peer review of "Biomaterials for Cell-Surface Engineering and Their Efficacy"

_jfb, 2021, doi:10.3390/jfb12030041_

Round 1

Reviewer 1 Report

The manuscript reviewed mammalian cell encapsulation and discussed the biomaterials employed for it and target efficacy.  Biomaterials play an essential role in cell encapsulation, which improves cell survival and regeneration activity in stem cell therapy. Thus, the topic is of significance.

I would have some comments.

  1. I would recommend briefly state the significance of stem cell therapy at the beginning of the Introduction, followed by the current issue that stem cell therapy has.
  2. Cell encapsulation technologies were developed, starting with microbial cells such as yeast. The readers might also be curious about their applications. Please briefly make a statement with the literature cited.
  3. Need ref cited: Line 37-40“Matrix molecules incorporated in the encapsulation hydrogel binds to matrix receptor on cell plasma membrane surface and activate signal transduction pathway leading to Akt protein phosphorylation – the marker for cell survival activation – which exert inhibitory effect on apoptosis induction.” Also, please correct some errors in English syntax, “------which exerts an inhibitory effect------.”
  4. In addition to citing them in Table 1, literature is also cited in the appropriate context.
  5. Please clarify the statement on lines 140-141, “However, those types of thick encapsulation exclude the possibility that the encapsulated cells be administered through IV injection because of vein occlusion.”
  6. Lines 143-147, “On the other hand, BM-MSCs subjected to surface modification by LbL assembly resulted in generation of 20 nm-thick non-continuous films around the plasma membrane surface. Although the LbL assembly did not significantly increase the cell size, BM-MSCs with nano-thin films showed significantly higher cell survival and recruitment to the intended injury site [46, 2].” It is hard to follow. Please clarify (1) BM-MSCs with nano-thin films showed significantly higher cell survival and recruitment to the intended injury site. I wonder how to explain the 20 nm-thick non-continuous films around the plasma membrane surface caused by LbL assembly? (2) Why did the author mention the cell size? In other words, is there any relationship between thick, thin films around the plasma membrane and the cell size in terms of LbL assembly?

      Also, please correct the English errors, “------the generation of 20 nm-thick------”, and “------nano-thin------.”

  1. Please spell out the abbreviation when first shown. For example:

Table 1, “BM-MSCs”;

Line 15, “ECM”;

Line 43, “MSC,” etc.

  1. Please correct the English typo. For example:

Line 34, no k in characteristicks.

Lines 180-181, “endothelial progenitor cells (EPCs)” instead of EPSs;

  1. The manuscript contains numerous errors in English syntax and typography. I recommend that the authors have their manuscript been reviewed by a native speaker or technical editor. For example:

Lines 119-122, “------ through a contact-independent mechanism [38], still significant number of cells ------”

Line 126, “------the end-organ------”;

Line 127, Add a comma, “------, attenuate------”

Line 129, please make sure “------recovery------” instead of “------recover------”?

Author Response

>>> We would like to appreciate the reviewers for the comments. We have revised our manuscript according to the comments, which helped our manuscript improved.

The changes made in the manuscript were highlighted by ‘track changes tool’.

The following is point-by-point responses to the reviewers’ comments.

Our point-by-point response are in blue color (in word file) and starts with ‘>>’.

Reviewer 1

The manuscript reviewed mammalian cell encapsulation and discussed the biomaterials employed for it and target efficacy.  Biomaterials play an essential role in cell encapsulation, which improves cell survival and regeneration activity in stem cell therapy. Thus, the topic is of significance.

>> Thank you for thorough reading of our manuscript and the positive comment!

I would have some comments.

  1. I would recommend briefly state the significance of stem cell therapy at the beginning of the Introduction, followed by the current issue that stem cell therapy has.

>> We added a paragraph commenting the significance of cell therapy and issues on Line23-30 in Page 1 as follows;

After finding various regenerative cell sources such as stem cells can exert essential functions to promote regeneration of tissues damaged by injury or disease, number of clinical applications of cell therapeutic agents is keep increasing. Although many cell-based therapies can provide valuable functionality that cannot be exerted by any other therapeutic options, duration of the efficacy could be limited due to the fact that in vivo environment is not favorable for survival of the transplanted cells. Cell encapsulation can increase the duration of cell survival by providing protection from the un-favorable environment.

  1. Cell encapsulation technologies were developed, starting with microbial cells such as yeast. The readers might also be curious about their applications. Please briefly make a statement with the literature cited.

>> We added a paragraph of brief description on starting of cell encapsulation with microbial cells on Line31-37 in Page 1 as follows;

Actually, cell-surface modification or encapsulation technology was originally developed with microbes such as yeasts [1] or Escherichia coli [2]. Various materials such as silica gel [3], poly(styrenesulfonate sodium salt), poly(allylamine hydrochloride) [1], chi-tosan, alginate, hyaluronic acid [2], poly(diallyldimethylammonium chloride) [4], silk [5] was adopted for encapsulation of microbes. However, to apply the cell encapsulation technology for mammalian cells, biocompatibility of the encapsulation process and its ef-fect on cell characterization should be considered.

  1. Need ref cited: Line 37-40“Matrix molecules incorporated in the encapsulation hydrogel binds to matrix receptor on cell plasma membrane surface and activate signal transduction pathway leading to Akt protein phosphorylation – the marker for cell survival activation – which exert inhibitory effect on apoptosis induction.” Also, please correct some errors in English syntax, “------which exerts an inhibitory effect------.”

>> Errors were corrected and the references were added.

  1. In addition to citing them in Table 1, literature is also cited in the appropriate context.

>> Many references were already in the main text, and we additionally added at the matching contents.

  1. Please clarify the statement on lines 140-141, “However, those types of thick encapsulation exclude the possibility that the encapsulated cells be administered through IV injection because of vein occlusion.”

>> Thank you for pointing out the jump of logic. We changed the description as follows;

However, thick encapsulation wall results in increased diameter of the encapsulated cell composite and increase the risk of vein occlusion upon IV injection, thereby excluding the possibility of systemic injection.

  1. Lines 143-147, “On the other hand, BM-MSCs subjected to surface modification by LbL assembly resulted in generation of 20 nm-thick non-continuous films around the plasma membrane surface. Although the LbL assembly did not significantly increase the cell size, BM-MSCs with nano-thin films showed significantly higher cell survival and recruitment to the intended injury site [46, 2].” It is hard to follow.
    Please clarify (1) BM-MSCs with nano-thin films showed significantly higher cell survival and recruitment to the intended injury site. I wonder how to explain the 20 nm-thick non-continuous films around the plasma membrane surface caused by LbL assembly?

>> Our description was somewhat obscure and too brief. We changed the description as follows;

Surface modification of BM-MSCs by LbL assembly using PLL/HA resulted in nano-thin films around the cell plasma membrane surface. LbL-assembled layers resulted in 20 nm-thick discontinuous and patch-like structure. Although this structure seemed not sturdy, both in vitro and in vivo cell survival was significantly potentiated and better recruitment to wound site was observed in muscle injury animal model [7,9,55].

(2) Why did the author mention the cell size? In other words, is there any relationship between thick, thin films around the plasma membrane and the cell size in terms of LbL assembly?

      Also, please correct the English errors, “------the generation of 20 nm-thick------”, and “------nano-thin------.”

>> You are right about point that cell surface modification do not influence the cell size. We corrected mistakenly described words “cell size” to “diameter of the encapsulated cell composite”.

  1. Please spell out the abbreviation when first shown. For example:

Table 1, “BM-MSCs”;

Line 15, “ECM”;

Line 43, “MSC,” etc.

>> We added full names.

  1. Please correct the English typo. For example:

Line 34, no k in characteristicks.

Lines 180-181, “endothelial progenitor cells (EPCs)” instead of EPSs;

>> Thank you for pointing out. We corrected the mistakes.

  1. The manuscript contains numerous errors in English syntax and typography. I recommend that the authors have their manuscript been reviewed by a native speaker or technical editor. For example:

Lines 119-122, “------ through a contact-independent mechanism [38], still significant number of cells ------”

Line 126, “------the end-organ------”;

Line 127, Add a comma, “------, attenuate------”

Line 129, please make sure “------recovery------” instead of “------recover------”?

>> Thank you for pointing out. We thoroughly read again, and further corrected the errors.

Reviewer 2 Report

This review is a high-quality manuscript on the Biomaterials for Cell Surface Engineering (especially for stem cells). The key materials and included and discussed. I have limited comments that if addressed the manuscript can be accepted. 

1) Please include the following references and discuss them.

Cell surface engineering and application in cell delivery to heart diseases, in Journal of Biological Engineering, Lee D.Y., et al., 2018.

Surface Engineering for Cell-Based Therapies: Techniques for Manipulating Mammalian Cell Surfaces, in ACS Biomater. Sci. Eng. 2018, 4, 11, 3658–3677, Abbina et al.

Potential of Cell Surface Engineering with Biocompatible Polymers for Biomedical Applications, in Langmuir 2020, 36, 41, 12088–12106, Teramura et al. 

2) Small discussion (e.g., one paragraph) on mechanical enhancement of cells will add value to the manuscript.

3) Minor comments: Page 2, line 62, delete "varies".

Author Response

We would like to appreciate the reviewers for the comments. We have revised our manuscript according to the comments, which helped our manuscript improved.

The changes made in the manuscript were highlighted by ‘track changes tool’.

The following is point-by-point responses to the reviewers’ comments.

Our point-by-point response are in blue color (in word file) and starts with ‘>>’.

Reviewer 2

This review is a high-quality manuscript on the Biomaterials for Cell Surface Engineering (especially for stem cells). The key materials and included and discussed. I have limited comments that if addressed the manuscript can be accepted. 

>> Thank you for the positive comment!

1) Please include the following references and discuss them.

Cell surface engineering and application in cell delivery to heart diseases, in Journal of Biological Engineering, Lee D.Y., et al., 2018.

Surface Engineering for Cell-Based Therapies: Techniques for Manipulating Mammalian Cell Surfaces, in ACS Biomater. Sci. Eng. 2018, 4, 11, 3658–3677, Abbina et al.

Potential of Cell Surface Engineering with Biocompatible Polymers for Biomedical Applications, in Langmuir 2020, 36, 41, 12088–12106, Teramura et al. 

>> While our review covers literatures for mammalian cell encapsulation for possible application on regenerative therapy, those review papers you suggested covers broader range of cell surface engineering. It is nice to remind that there are diverse methods adopted for cell surface engineering. Therefore, we added a reminding paragraph about diverse technic citing the literatures, and commented that this review is focused on non-covalent modification of mammalian cell encapsulation at the end of Introduction section as follows;

As much as the subjected cells are varied, the methods of cell surface engineering are diverse. Functional small molecules could be tethered on cell surface by covalent conjugation methods such as chemical conjugation, enzymatic/metabolic conjugation, physical conjugation, or UV-mediated grafting. There are several review papers covering diverse techniques adopted for cell surface engineering [12-14].

2) Small discussion (e.g., one paragraph) on mechanical enhancement of cells will add value to the manuscript.

>> Thank you for pointing out. We realized that we didn’t add enough description on cell survival due to improved mechanical strength. We corrected previous writing and added more description as follows;

The high susceptibility to mechanical stress is based on the fact that lipid bilayer of cell plasma membrane is constructed by hydrophobic interaction between phospholipid molecules. However, hepatocytes, highly susceptible to mechanical stress in attachment-deprived state, show mechanical stability inside the liver tissue surrounded by matrix molecules. Likewise, cell surface modification by deposition of hydrogel or matrix molecules can dramatically reduce mechanical insult by providing mechanical stability. When HepG2 cells (hepatocyte carcinoma) were coated with FN-gelatin or Col IV-LN by LbL assembly, more than 85% of the cells survived after 9 rounds of centrifugation while non-coated control group showed poor cell survival (6%) [8]. Furthermore, application of high-mechanical strength molecules can enhance mechanical stability even further. In a study reported by Cha C, cells seeded on microcarrier-like gelatin core were encapsulated with silica. The silica encapsulation protected the cells form mechanical stress and resulted in significantly increased viability [40]. Systemic transplantation to in vivo animal model also proved that surface-modified cells with LbL assembly resulted in better cell survival [7,15,69].

3) Minor comments: Page 2, line 62, delete "varies".

>> Thank you for pointing out. We corrected the mistake.

Round 2

Reviewer 1 Report

In my opinion, it can be accepted for publication after moderate English changes.

For example:

  1. Lines 23-30,
    • Add an article, (the) number of clinical applications of cell therapeutic agents.
    • (Keeps) increasing.
    • (the) duration of the efficacy
    • by (protecting) from the (unfavorable) environment.
  2. Lines 31-37,
    • were adopted for encapsulation of microbes (The singular verb does not appear to agree with the plural subject materials).
    • styrene sulfonate (correct your spelling).
    • chitosan (correct your spelling).
    • diallyl dimethylammonium chloride (correct your spelling).
    • the biocompatibility of the encapsulation process (Add an article).
  3. Lines 154-156,
    • However, (a) thick encapsulation wall results in (an) increased diameter of the encapsulated cell composite. (It) increase(s) the risk of vein occlusion upon IV injection, thereby excluding the possibility of systemic injection.

etc.

Author Response

>>> We thoroughly checked for English errors and added figures with permission from previous literatures because words without data could sound obscure.

The changes made in the manuscript were highlighted by ‘track changes tool’.

The following is point-by-point responses to the reviewers’ comments.

Our point-by-point response are in blue color (in word file) and starts with ‘>>>’.

In my opinion, it can be accepted for publication after moderate English changes.

For example:

  1. Lines 23-30,
    • Add an article, (the) number of clinical applications of cell therapeutic agents.
    • (Keeps) increasing.
    • (the) duration of the efficacy
    • by (protecting) from the (unfavorable) environment.
  2. Lines 31-37,
    • were adopted for encapsulation of microbes (The singular verb does not appear to agree with the plural subject materials).
    • styrene sulfonate (correct your spelling).
    • chitosan (correct your spelling).
    • diallyl dimethylammonium chloride (correct your spelling).
    • the biocompatibility of the encapsulation process (Add an article).
  3. Lines 154-156,
    • However, (a) thick encapsulation wall results in (an) increased diameter of the encapsulated cell composite. (It) increase(s) the risk of vein occlusion upon IV injection, thereby excluding the possibility of systemic injection.

etc.

>>> Thank you for your meticulous review. We thoroughly checked for English errors throughout the manuscript.